# Climate Change Perception and Vulnerability Assessment of the Farming Communities in the Southwest Parts of Ethiopia

**Dessalegn Obsi Gemeda** [1,*] , **Diriba Korecha** [2] **and Weyessa Garedew** [3]

1   Department of Natural Resources Management, Jimma University College of Agriculture and Veterinary Medicine, Jimma University, Jimma P.O. Box 307, Ethiopia
2   Famine Early Warning Systems Network, Addis Ababa P.O. Box 17403, Ethiopia; dkorecha@fews.net
3   Department of Horticulture and Plant Sciences, Jimma University College of Agriculture and Veterinary Medicine, Jimma University, Jimma P.O. Box 307, Ethiopia; woyessa.garedew@ju.edu.et
*   Correspondence: dasoobsi@gmail.com or dessalegn.obsi@ju.edu.et

**Abstract:** This study assesses the perceptions and vulnerability of the farming communities to climate change in the southwestern parts of Ethiopia. Climate change vulnerability assessment is a prerequisite to designing climate change adaptation strategies. A multistage cluster sampling technique was used to select four of the six zones from the southwestern parts of Oromia. Close-ended and open-ended questionnaires were used to assess household perceptions of climate change and the degree of vulnerability to climate change by using five household capitals: natural, social, financial, physical, and human capital. Data were collected from 442 households in 4 districts: Jimma Arjo, Bako Tibe, Chewaka, and Sekoru. The vulnerability of the farming communities was assessed using the households' livelihood vulnerability index. A total of forty indicators from five capitals were applied to calculate household livelihood vulnerability to climate change. Household perceptions of climate change had a statistically significant relationship with changes in rainfall pattern (75.6%, $p < 0.001$), temperature pattern (69.7%, $p < 0.001$), drought (41.6%, $p = 0.016$), flood (44.1%, $p = 0.000$), and occurrence of early (53.2%, $p < 0.001$) and late rain (55.9%, $p < 0.001$). The results show that households in the Sekoru district were the most vulnerable (0.61), while households in the Jimma Arjo district were less vulnerable (0.47) to the effect of climate change. Household vulnerability to climate change is mainly related to the occurrence of drought, lack of much-needed infrastructure facilities, and weak institutional support. Links with financial organizations are also lacking in the household. The findings of this study will help policymakers to address the impact of climate change. To support disaster risk management on the one hand and increase the resilience of vulnerable societies to climate change on the other, we recommend a detailed assessment of the remaining districts of the region.

**Keywords:** Southwest Ethiopia; farming communities; climate change; perception; vulnerability; capital; livelihood vulnerability index





## 1. Introduction

The global mean temperature is increasing, reducing agricultural yield and threatening food security and people's livelihoods [1,2]. Global climate change is also increasing the occurrence of drought, floods, and other climate extremes. People experiencing extremes, mainly drought and floods, can influence their perceptions of climate change. Declining agricultural yields expose farming communities to food insecurity and malnutrition; when the agricultural system is exposed to climate extremes, it reduces yield production and aggravates community vulnerability. While having adaptive capacity, the vulnerability of the communities to climate change will be minimized. It is, therefore, crucial to assess household perceptions of climate change and the level of vulnerability of farming communities to inform decision makers to design effective climate change adaptation strategies. Investigating the vulnerability of rainfed-dependent agriculture to climate change impacts

is highly recommended to inform policymakers to design adaptation strategies at different scales. Households' views and perceptions of climate change directly depend on their understanding, experience, and beliefs about climate change issues. Farming communities need to be aware of climate change-related problems to take adaptation measures. People who are unaware of the impacts of climate change are reluctant to adapt and cope with the consequences of climate change. A clear understanding and belief in the consequences of climate change could be a key determinant of adaptation action.

Household perception of climate change is one of the main elements that can enhance the adaptation process. Farmers who perceive the impacts of climate change are more likely to use various climate change adaptation options to minimize their vulnerability [3]. Developing countries are more vulnerable to the effects of climate change than developed countries due to their financial and technical weaknesses and low capacity to adapt [4,5]. In contrast, developed nations generally have a low degree of vulnerability and a high degree of adaptive capacity, which is a function of natural, technological, human, financial, and social capital [6].

Several studies have shown that Africa is more exposed to climate change than other continents due to its heavy reliance on rain-fed agriculture and limited adaptive capacity [7–12]. Similarly, in South Asian countries such as Pakistan, the impacts of climate change are critical as a large proportion of the farming communities depend on rain-fed agriculture [13,14]. Compared to other continents, Africa is the most vulnerable to climate change [15]; however, it has the lowest GHG emissions [7]. The Eastern and Western African countries are projected to be most affected by climate change [16].

Like other African countries, the farming communities in southwestern Ethiopia are vulnerable to climate change due to heavy dependence on agriculture, which is climate sensitive. A study by [17] indicated that the amount of rainfall in southwestern Ethiopia is inconsistent, and some stations even experience a declining trend during the crop-growing season. Statistically significant increasing trends of mean maximum and minimum temperature are reported in the study area [17]. The declining trends of rainfall during the crop growing seasons and significant increasing trends of temperature driven by land use and land cover change can negatively affect community livelihoods, which can expose the community to food insecurity and poverty. The extent of climate change vulnerability varies across regions, economic sectors, and social groups. Climate change has an enormous impact on poor, young, elderly, and marginalized people because of their poor adaptive capacity [18–20]. Some social groups within the same livelihood system have various capacities to minimize the effects of climate change. Poor households are the most at risk of climate change due to a lack of access to risk management [21].

Vulnerability is the outcome of high susceptibility to harm and a weak capacity to cope and adapt [20]. It is the degree to which a system is susceptible to or unable to cope with climate change impacts [22]. Vulnerability to climate change is a function of exposure, sensitivity, and adaptive capacity [23–30]. It is also positively correlated with exposure and sensitivity and has a negative relationship with adaptive capacity; that is, the higher the exposure and sensitivity are, the more vulnerable, while the higher the adaptive capacity is, the less vulnerable [26,31,32].

Vulnerability assessment is a prerequisite to designing climate change adaptation strategies [33–35]. To date, various techniques have been used to assess community vulnerability to climate change. For instance, three indicators, namely exposure, sensitivity, and adaptive capacity, have been used to measure community vulnerability to climate change [26,32,36–38]. Econometric and indicator-based methods have also been used; the econometric method uses household-level socioeconomic survey data [39], while the indicator-based method systematically combines natural, social, financial, physical, and human capital to measure vulnerability status [23,26,30,40–45]. This study used an indicator-based approach, which is the most common method of demonstrating the power of each factor in vulnerability assessment [46,47].

Several climate change trend assessments have been conducted in southwestern Ethiopia [48–53]. Although climate change trend assessments have been conducted by different scientists in the past, the vulnerability of households to climate change in southwestern Ethiopia has received less attention. Research on climate change perception in the southwestern part of Ethiopia is still limited. Although the impacts of climate change have already been reported by various scientists in southwestern Ethiopia, climate change perception and vulnerability assessments are not well documented, especially in southwestern Ethiopia. In line with this, Ethiopia's Climate Resilient Green Economy Strategy claims a lack of climate change vulnerability assessment, monitoring, and climate adaptation mainstreaming [54]. This study, therefore, aims to address the existing research and knowledge gaps on community vulnerability to climate change in southwestern Ethiopia. Moreover, this study is more comprehensive and includes natural, social, financial, physical, and human capital to assess household vulnerability to climate change.

This study is structured into four sections. The first section (Introduction) introduces global climate change and its impacts, with special emphasis on developing countries, which are the most vulnerable due to heavy dependence on rain-fed agriculture. The research gaps and aim of the study are also introduced in this section. The next section (Section 2) is dedicated to materials and methods: study area descriptions, study design, sampling procedure and sample size, techniques adopted to analyze farmers' perception of climate change, and livelihood vulnerability analysis. This is followed by results and discussion (Section 3). The final section (Section 4) is dedicated to conclusions. Lastly, we provide all cited references in this study.

## 2. Materials and Methods

### 2.1. Study Area Descriptions

This study was conducted in four Zones [West Shewa, Buno Bedele, East Wollega, and Jimma] from western parts of Oromia (Figure 1). Four districts, namely Sekoru, Chewaka, Jimma Arjo, and Bako Tibe, were purposively selected in this study. All districts are located in the southwestern part of Oromia. The economy of the study area relies heavily on rain-fed agriculture, which is the most vulnerable to climate change. A summary of the study area, including the location, population, topography, climate, and especially rainfall and temperature of each district, is provided below.

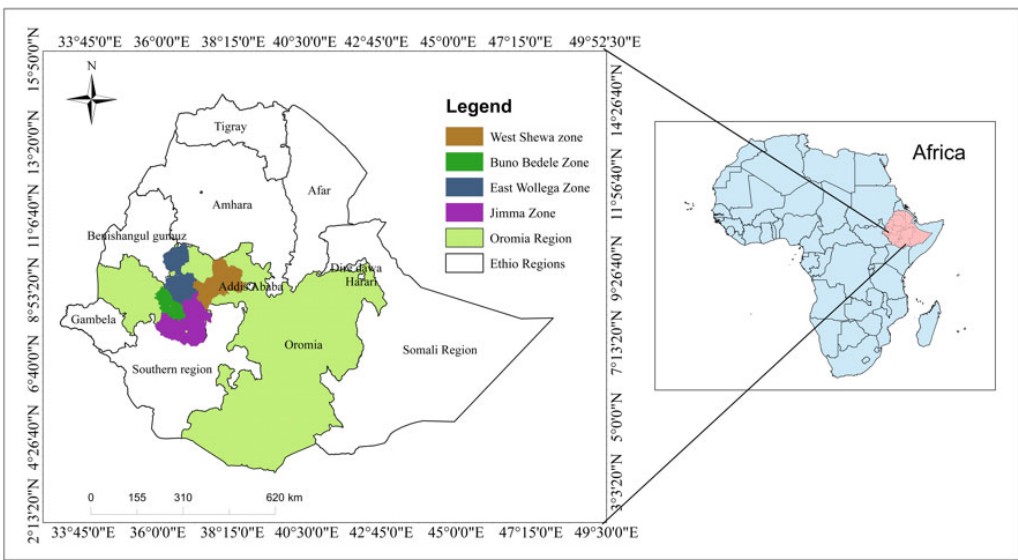

**Figure 1.** Map of the study area.

Sekoru is one of the 20 districts in Jimma Zone. The district lies between 7.55° N and 7.92° N, and 37.25° E to 37.42° E. The district has a total population of 136,320, of

which 68,469 and 67,851 are males and females, respectively, as documented by the Central Statistical Agency of Ethiopia [55]. The average annual rainfall is approximately 1360 mm, with mean annual minimum and maximum temperatures of 13.3 °C and 26.2 °C, respectively [56].

Chewaka, which is one of the districts in the Buno Bedele zone, is located between 80.43° N and 9.50° N, and 35.58° E and 36.14° E. The district has a total of 28 villages (*Kebeles*) with an estimated population of 75,111, and 15,649 households [57]. The annual rainfall ranges from 800 to 1200 mm, and the mean temperature varies between 19.8 °C and 28.5 °C. Chewaka is the largest resettlement area in the southwestern parts of Ethiopia [58]. Maize, sorghum, rice, sesame, and soybean are the most stable crops.

Jimma Arjo district is located in the southwestern part of the East Wollega Zone and situated between 8.22° N and 8.55° N, and 36.20° E and 36.41° E, with a total area of 773 km². The district has a population of 86,329, of whom 42,093 are male, and 44,236 are female [55]. This district is characterized by a humid tropical climate and receives a mean annual rainfall of 1702 mm with mean minimum and maximum temperature variations between 11.2 °C and 13.2 °C, and 23.8 °C and 25 °C, respectively [17].

Bako Tibe district is located in the West Shewa Zone and is situated between 8.55° N and 9.14° N and 37.01° E and 37.17° E. The district has a total population of 123,031, of which 61,018 and 62,013 are males and females, respectively [55]. The district receives maximum rain from June to September and an average annual rainfall of 1006 mm, with mean minimum and maximum temperatures between 12.9 °C and 28 °C, respectively [17]. Teff, maize, and wheat are the main cereal crops grown in this area. The key informant interviews highlight that climate change in the study area disturbs the normal crop calendar. This is due to the occurrence of early and late rains. Due to the increasing trends of temperature, some crops increase yields while others experience declining trends, as different crops require different optimum temperatures. The increasing trends of crop diseases and pests are also associated with climate change.

## 2.2. Study Design

A mix of quantitative and qualitative research designs [59,60] were used to assess the vulnerability of farming communities to climate change. Structured questionnaires (close-ended and open-ended questions) were used to assess household perceptions of climate change and the degree of vulnerability to climate change based on five household capitals: natural, social, financial, physical, and human capital [27,61]. Five types of capital, including demographic, educational status, climate change and variability, income status of the household, and accessibility to different services and infrastructures, are included based on the literature and key informant interviews. All influencing factors were combined, depending on their association, into five types of capital, with twelve sub-indicators for natural capital, seven for social capital, seven for financial capital, eight for physical capital, and six for human capital. A total of 40 indicators were selected through stakeholder consultations and interviews with key informants. After the identification of all subcomponents, equal values were given (normalization as zero and one). A questionnaire-based survey was conducted with 442 randomly selected households from the study area's districts and villages. A conceptual framework developed for this study is presented (Figure 2). It is important to indicate the relationship between climate change and the five capitals that can determine the level of vulnerability to climate change. Community vulnerability is the function of exposure, sensitivity, and adaptive capacity [27,30]. The presence of the five capitals allows the public to use different strategies to minimize the extent of vulnerability to the impacts of climate change. In contrast, weakness in human, natural, social, financial, and physical capital contributes to vulnerability to climate change, leading to poverty, and setbacks economic development of a country. Similarly, climate variability and the occurrence of extreme climates, such as droughts and floods, can reduce the adaptive capacity of communities to the impacts of climate change.



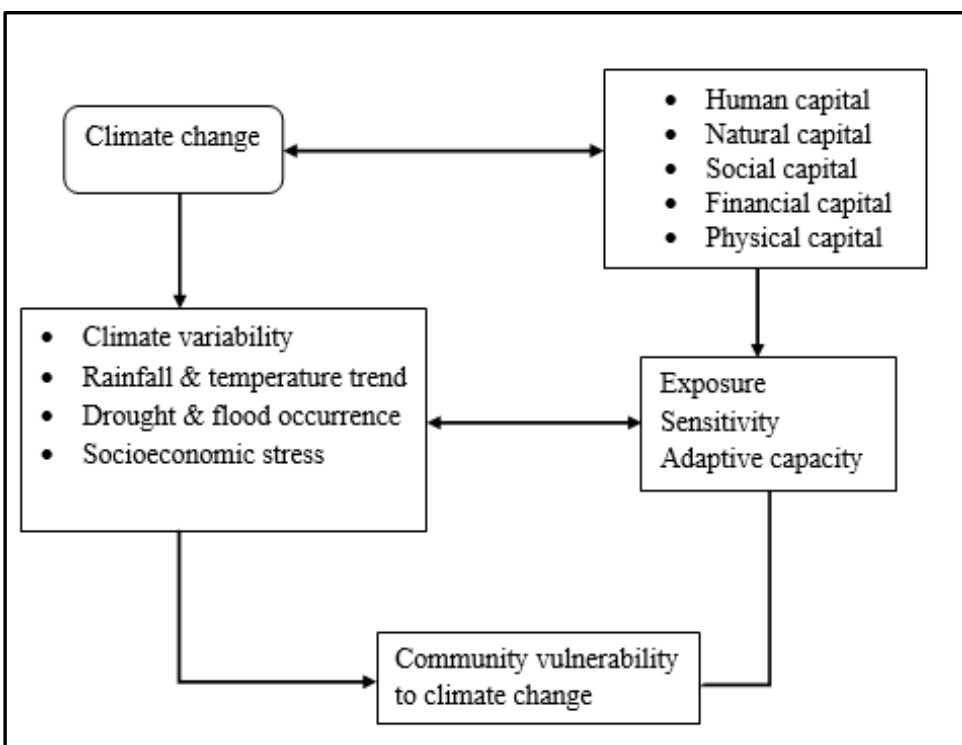

**Figure 2.** Conceptual framework of the study.

### 2.2.1. Sampling Procedure and Sample Size

A multistage cluster sampling technique was used to select four of the six zones from the southwestern parts of Oromia. Thus, Sekoru, Chewaka, Jimma Arjo and Bako Tibe districts were selected from Jimma, Buno Bedele, East Wollega, and West Shewa Zones, respectively. Next, four districts, Sekoru (Jimma), Jimma Arjo (East Wollega), Chewaka (Buno Bedele), and Bako Tibe (West Shewa), were selected in consultation with stakeholders based on community exposure to climate change. We also take into account the reliance of the farming communities on rain-fed agriculture in the area, which is one of the most vulnerable to climate change and variability. After the study districts were identified, key informant interviews with district agricultural experts with relevant knowledge on climate change and then sample villages were selected for household interviews. Finally, four villages, namely, Abelti in Sekoru (N = 84), Gudure in Chewaka (N = 147), Hare in Jimma Arjo (N = 121), and Oda Gibe in Bako Tibe (N = 90), were selected for data collection. Following a multistage sampling procedure, 442 household heads were interviewed from the selected four districts.

The main criteria for choosing study sites were the presence of meteorological stations, the variability of rainfall, the occurrence of climatic extremes (such as excessive precipitation and increasing temperatures), changes in the climate suitability of some crops, and stakeholder recommendations. In addition to the exposure of communities to climate change, we also take into account the presence of long historical weather stations, agroecological zones, and topographic variation. The selected sites have significant topographical variation; thus, the elevations are Jimma Arjo (1280.67 to 2563.77 m), Bako Tibe (900 to 1281 m), Chewaka (1130–2053 m), and Sekoru (1300 to 1800 m). After study villages were identified, a proportional sampling method was used, and a total of four hundred forty-two (442) households were sampled using the technique developed by [62] to determine the sample size at the 95% confidence level. Accordingly, there were a total of 147 in Chewaka, 121 in Jimma Arjo, 84 in Sekoru, and 90 in Bako Tibe.

In this study, the method of triangulation proposed by Jick [63] and used by several authors [64–67] was adopted to use multiple techniques that utilize both quantitative and qualitative data. Rossman and Wilson [64] demonstrated that triangulation tech-

niques allow researchers to rely on multiple types of data to enhance the accuracy of their conclusions. Triangulation methods allow us to integrate the reliability and validity of meteorological data outputs with community perceptions of climate change and community exposure to climate change impacts. The Delphi method [68–71] was also adopted to design the questionnaire from eight key informants on community vulnerabilities, adaptation strategies, and the existing adaptation barriers.

### 2.2.2. Household Perceptions of Climate Change

A household survey was conducted to assess the farmers' perceptions of climate change and the extent of household vulnerability to this change. This study adopted a binary logistic model [72], which uses a binary-based response; that is, the value of one (1) indicates the probability of perceiving climate change, and zero (0) if otherwise [36,73,74].

### 2.2.3. Livelihood Vulnerability Index Analysis

The livelihood vulnerability index was calculated using five types of capital: natural, social, financial, physical, and human. Each capital/asset is standardized as an index, as recommended by many authors [23,26,30,36]. After the standardization of each indicator, the subcomponents were averaged [27,38] using Equation (1).

$$M_v = \frac{\sum_{i=1}^{5} Index_{sv}}{n} \tag{1}$$

where $M_v$ is the average index value of one major component, $Index_{sv}$ is the index value of each indicator for the respective major components of vulnerability, and n is the number of indicators for each major vulnerability component.

Indictor-based climate vulnerability assessment was employed by creating a single indicator composite index [24] and normalizing them (zero and one). First, the household livelihood vulnerability index (*HLVI*) was applied to assess livelihood vulnerability to climate change [20,28,34,38,75,76]. Then, once equal values for the five major components of a district were obtained, the overall *HLVI* was calculated based on five major capitals, i.e., natural (*N*), social (*S*), financial (*F*), physical (*P*), and human (*H*) capital using (Equation (2)).

$$HLVI = \frac{W_{e1N} + W_{e2S} + W_{e3F} + W_{e4P} + W_{e5H}}{W_{e1} + W_{e2} + W_{e3} + W_{e4} + W_{e5}} \tag{2}$$

where *HLVI* is the household livelihood vulnerability index, while $W_{e1}$, $W_{e2}$, $W_{e3}$, $W_{e4}$, and $W_{e5}$ are the weights of indicators for natural (*N*), social (*S*), financial (*F*), physical (*P*), and human (H) capital, respectively.

For this study, $W_{ei} = 1$ for all *i* due to the simplicity and uniform importance of the five capitals. The five livelihood assets are equally important in household vulnerability analysis. Each of the five capitals has different sub-components: natural capital (12 sub-indicators), social capital (7 sub-indicators), financial capital (7 sub-indicators), physical capital (8 sub-indicators) and human capital (6 sub-indicators). Equal values were assigned [77–80] to all sub-components assuming that all contribute to vulnerability to climate change. The higher the value of *HLVI*, the more vulnerable, while the lower the value, the less vulnerable [26]. In this study, household vulnerability index score values near one indicate high vulnerability, while values near zero indicate high resilience.

## 3. Results and Discussion

### 3.1. Sociodemographic Variables

The results of the sociodemographic characteristics of the respondents showed that there were 359 (81.2%) male-headed households out of the 442 household heads, which was almost five times greater than that of female-headed households. Previous studies have shown that male-headed households are more likely to implement climate change adaptation strategies than female-headed households [81–83]. The lower representation

of female household heads in the study area was related to cultural patterns. In terms of household age distribution, nearly 13% ranged from 20 to 30.

Regarding marital status, most of the households' heads were married (85.1%), while approximately 8.1, 3.6 and 3.2% were widowed, divorced, and single, respectively. The majority of the household heads, 235, were illiterate (53.2%), while 207 (46.8%) were literate. Of the total literate household heads, 207 (46.8%), 178 (40.3%), and 29 (6.6%) had attained primary and secondary school, respectively. It is clear that educated families can easily evaluate the effect of climate change on their livelihoods and have a major influence on taking appropriate adaptation strategies. Education can enhance individual knowledge [84], which increases resilience to climate shocks. Studies show that there is a positive correlation between education and farmers' willingness to adopt an adaptation strategy to climate change impact [83,85,86].

Households aged between 31 and 40, 41 and 50, and above 51 accounted for 29.2, 25.3, and 32.8% of households, respectively. Regarding religious affiliation, Islam is the dominant religion in the sample households, with a share of 49.3%, followed by Protestant (35.1%) and Orthodox Tewahedo, with a share of approximately 15.6%. In addition, most of the households had large family sizes. Accordingly, 47.3% of the households had a family size greater than 7, which is greater than the national average family size of 4.9 [87].

Approximately 41% and 11.8% of the household heads had family sizes of 4–6 and 1–3, respectively. The age structure of the household heads in the study area indicates that approximately 42% of the population was concentrated under the age of 15 years, with older age (>65 years) being small (4%). Age composition has a strong influence on the food security of the household. Economically active age groups (15–64) accounted for 54% of the sampled household heads. The sampled household age dependency ratio was 0.87 (87%), which exceeds the country age dependency ratio of approximately 0.77 (77%) [88].

### 3.2. Farming Communities' Perceptions of Climate Change

Because of an increase in temperature and rainfall fluctuations in the study area, the majority of the households (323 out of 442) perceive climate change. It is unequivocal that climate change is occurring in every country across seven continents due to the overexploitation of natural resources, leading to global warming trends. The results show that there is a significant relationship between climate change perceptions and changes in rainfall pattern ($p < 0.001$), change in temperature pattern ($p < 0.001$), drought occurrence ($p = 0.016$), recent drought occurrence ($p < 0.001$), recent flood occurrence ($p = 0.000$), flood frequency ($p = 0.009$), and the occurrence of early rain and late rain ($p < 0.001$). Most of the households perceive that there is a change in rainfall (75.6%) and a change in temperature patterns (69.7%). Thus, the farming communities have experienced changing rainfall patterns, increasing trends of temperature and rainfall irregularities, which have had an impact on people's livelihoods. The variables used to understand household perceptions of climate change are presented in (Table 1).

Although statistically significant results have been obtained on the occurrence of climate extremes such as droughts and floods, more than 50% of the sampled households do not perceive the occurrence of droughts and floods. For instance, the majority of the farming community (66.3%) in the study area did not perceive drought occurrence in recent decades. However, 33.7% said they had drought problems in the study area. This indicates that all people in the study area have different levels of understanding of climate change and associated problems. Some of the elders in the study area confirmed that rainfall is declining and that it may not rain at the right time to prepare the land for agriculture, which could affect the agricultural system. Farming communities claim that the beginning and end of the rainy season are often confusing and different from normal conditions. Similarly, 55.9% and 56.8% of the respondents do not perceive recent floods and frequent flood occurrences, respectively. However, the results of key informant interviews indicate that extreme events such as droughts and floods have recently increased. Increases in the frequency and severity of droughts and floods are projected to affect sustainable

development [22]. The contrasting findings are because more than 50% of the households were illiterate and did not clearly elaborate on climate change, while the key informants had an analytical capacity to express their knowledge and experiences on drought and flood occurrence.

**Table 1.** Household perceptions of climate change.

| Indicators of Climate Change | Perceived | Not Perceived | Chi-Square | *p* Value |
|---|---|---|---|---|
| Change in rainfall pattern | 75.6 | 24.4 | 37.14 | <0.001 ** |
| Change in temperature pattern | 69.7 | 30.3 | 50.38 | <0.001 ** |
| Occurrence of drought events | 41.6 | 58.4 | 5.76 | 0.016 * |
| Recent drought occurrence | 33.7 | 66.3 | 17.83 | <0.001 ** |
| Recent flood occurrence | 44.1 | 55.9 | 13.48 | 0.000 ** |
| Recent flood frequency | 43.2 | 56.8 | 6.66 | 0.009 ** |
| Occurrence of early rain | 53.2 | 46.8 | 16.27 | <0.001 ** |
| Occurrence of late rain | 55.9 | 44.1 | 50.79 | <0.001 ** |
| Taking action against climate change | 43.2 | 56.8 | 44.81 | <0.001 ** |
| Crop loss due to rain deficit | 47.5 | 52.5 | 33.88 | <0.001 ** |
| Food insecurity due to climate change | 49.3 | 50.7 | 22.65 | <0.001 ** |
| Climate change affects human health | 47.7 | 52.3 | 17.18 | <0.001 ** |

Significance levels: * $p \leq 0.05$; ** $p \leq 0.01$. Note: the values in the raw are percentages based on the sample size of 442.

The farming communities perceived the occurrence of early rain (53.2%, $p < 0.001$) and late rain (56%, $p < 0.001$). Investigation of farmers' perceptions of climate change is a precondition for assessing adaptation strategies [14,83]. Rainfall irregularity is one of the key problems of the rain-fed dependent agricultural economy. High interannual variability in rainfall and temperature has been observed recently in the southwestern parts of Ethiopia [17,56].

Concerning the association between crop loss and food insecurity with climate change, the number of perceived respondents was comparable with those who did not perceive climate change ($p$-value < 0.001). Household nutrition and livelihoods are directly dependent on climatic factors [14]. Climate change is projected to cause a decline in cereal production in countries such as Ethiopia, where the majority of the people rely on rain-fed agriculture. The key informants occasionally recognized the occurrence of drought and floods affecting agricultural crops in the past. They also understand the increasing trend of extreme drought across the study area. A study by [89] indicated the preferences of the farming communities to use drought-resistant crops in southwestern parts of Ethiopia. Taking action on the adverse effects of climate change was another concern for the farming communities. The results revealed that approximately 43% ($p < 0.001$) took measures such as crop diversification, crop rotation, and the use of improved crop and livestock varieties, while approximately 57% did not take any actions against climate change effects. A study by [90] indicated that if people do not believe in the occurrence of climate change, they may not implement adaptation actions. Household heads who are aware of climate change grow multiple crops at once and alternately grow different crops to improve the soil's nutrients. According to an interview with key informants in the Jimma Arjo district, farmers grow Niger seeds and linseed when the soil becomes less fertile.

Farmers also grow crops such as peas and beans to increase soil fertility in the study area. Concerning human health issues, there was a significant relationship between climate change and human health (47.7%, $p < 0.001$). Climate change, particularly the increase in temperature in highland areas, likely increases the risk of malaria. A study on Sub-Saharan African countries revealed that malaria prevalence was significantly positively correlated with temperature and precipitation [91].

*3.3. Indicator of Household Vulnerability to Climate Change*

Compared to other sources of revenue, the livelihoods of the farming communities were the most vulnerable to climate change. Weak natural, social, financial, physical, and human capital increases the vulnerability of farming communities to the impacts of climate change. Bewket [18] highlighted that climate change is aggravating the problems of vulnerable and poor people in marginal areas. The problem of climate change in the developing world is worst due to poor capacity to combat climate change impact [13]. In contrast, access to natural, social, financial, physical, and human capital increased community resilience to climate change [41,92–94]. On the other hand, unequal access to resources, climate hazards, and food insecurity can aggravate community vulnerability to climate change [22].

3.3.1. Natural Capital

The first major component was natural capital, which was assessed by 12 indicators of the household vulnerability index, including the availability of fertile land, the existence of water resources for irrigation, grazing land, potable water, use of rivers and streams for drinking, climate suitability for agriculture, drought occurrence, flood hazards, exposure to cold temperatures, exposure to hot temperatures, and occurrence of late and early rain. Floods can ultimately damage the existing infrastructure and affect households' food security by destroying agricultural crops [95]. All 12 sub-indicators were given the same value and normalized (0 and 1).

The farming communities in the study area were vulnerable to climate change impacts. The existing physical capital in the study area is insufficient to fight the impacts of climate change. Poor landless households and large-sized families are most vulnerable to climate change [28]. When it comes to natural capital, there are significant differences between districts. For instance, the natural capital vulnerability index ranged from 0.33 for Jimma Arjo to 0.62, 0.63, and 0.68 for Bako Tibe, Sekoru, and Chewaka districts, respectively. This clearly shows that the Chewaka district has less natural capital than Sekoru, Bako Tibe and Jima Arjo districts (Table 2).

**Table 2.** Natural capital vulnerability index of Jimma Arjo, Bako Tibe, Chewaka, and Sekoru districts.

| Indicators of Household Vulnerability Index | Composite Index | | | |
|---|---|---|---|---|
| | Jimma Arjo | Bako Tibe | Chewaka | Sekoru |
| Availability of fertile land for agriculture | 0.45 | 0.62 | 0.76 | 0.94 |
| Existence of water resources for irrigation | 0.58 | 0.77 | 0.73 | 0.82 |
| Existence of grazing land for livestock | 0.19 | 0.76 | 0.68 | 0.83 |
| Potable water for household | 0.57 | 0.47 | 0.61 | 0.67 |
| Agricultural drought occurrence | 1.00 | 1.00 | 1.00 | 1.00 |
| Climate suitability for agricultural production | 0.09 | 0.22 | 0.57 | 0.23 |
| Rainfall deficit in the study area | 0.08 | 0.56 | 0.60 | 0.35 |
| Floods hazardous | 0.12 | 0.68 | 0.70 | 0.45 |
| Extreme cold occurrence | 0.08 | 0.46 | 0.70 | 0.69 |
| Exposures to extreme high temperature | 0.10 | 0.74 | 0.56 | 0.74 |
| Occurrence of late rain | 0.37 | 0.56 | 0.60 | 0.41 |
| Occurrence of early rain | 0.36 | 0.61 | 0.66 | 0.40 |
| Natural capital vulnerability index | 0.33 | 0.62 | 0.68 | 0.63 |

The high vulnerability of Chewaka district may be associated with agricultural drought (1.00), availability of fertile land for agriculture (0.76), the existence of water resources for irrigation (0.73), flood hazards (0.70), extreme cold (0.70), the existence of grazing land (0.68), and the occurrence of early rain (0.66) and late rain (0.60). Natural capital helps communities restore their former state when livelihoods face environmental challenges [43]. In addition, provisions of health and social security are required to reduce social vulnerability to climate change during and after floods [33].

Sekoru district was the second most vulnerable based on natural capital, which was connected with the occurrence of agricultural drought (1.00), fertile land (0.94), grazing land (0.83), water resources (0.82), high temperature (0.74), and extreme cold (0.69). Comparable results were found for Sekoru (0.63) and Bako Tibe (0.62). Poor rural households with limited land resources for agricultural production are vulnerable to climate change [28]. Moreover, infertile land and limited financial capital to afford chemical fertilizers are the main challenges facing households in adapting to the effects of climate change.

The scores of agricultural droughts (1.00), water resources (0.77), grazing land (0.76), extreme temperature (0.74), flooding hazard (0.68), fertile land (0.62), and early rain (0.61) were among the major driving forces for the natural vulnerability of Bako Tibe smallholder farmers. The key informant interviews highlight that increases in food shortages in the region are related to an increase in rainfall irregularities during the main growing season and an increase in climate extremes such as droughts and floods. Human-induced water scarcity is projected to increase in the future, leading to food insecurity [33]. The presence of fertile land (0.45), grazing land (0.19), climate suitability for agricultural production (0.09), exposure to floods (0.12), extreme temperature (0.10), extreme cold (0.08), and rainfall deficits (0.08) makes Jimma Arjo district less vulnerable than the other three districts.

### 3.3.2. Social Capital

The second major component was social capital, which was assessed using seven indicators (community-based organization membership, access to climate information, access to chemical fertilizers, linkage with financial institutions, access to government subsidies, access to disaster relief assistance, and obtaining loans without a contract from friends). All seven components are given equal values and normalized (0 and 1). The results of the social capital index score indicated that the Jimma Arjo district was less vulnerable (0.29), while Chewaka (0.42) was moderately vulnerable compared to the Bako Tibe and Sekoru districts (0.55). In the Jimma Arjo district, among the seven subcomponents of social vulnerability indicators, access to government subsidies (0.63), and disaster relief assistance (0.51) are the two major factors that influenced the vulnerability of the farming communities (Figure 3).

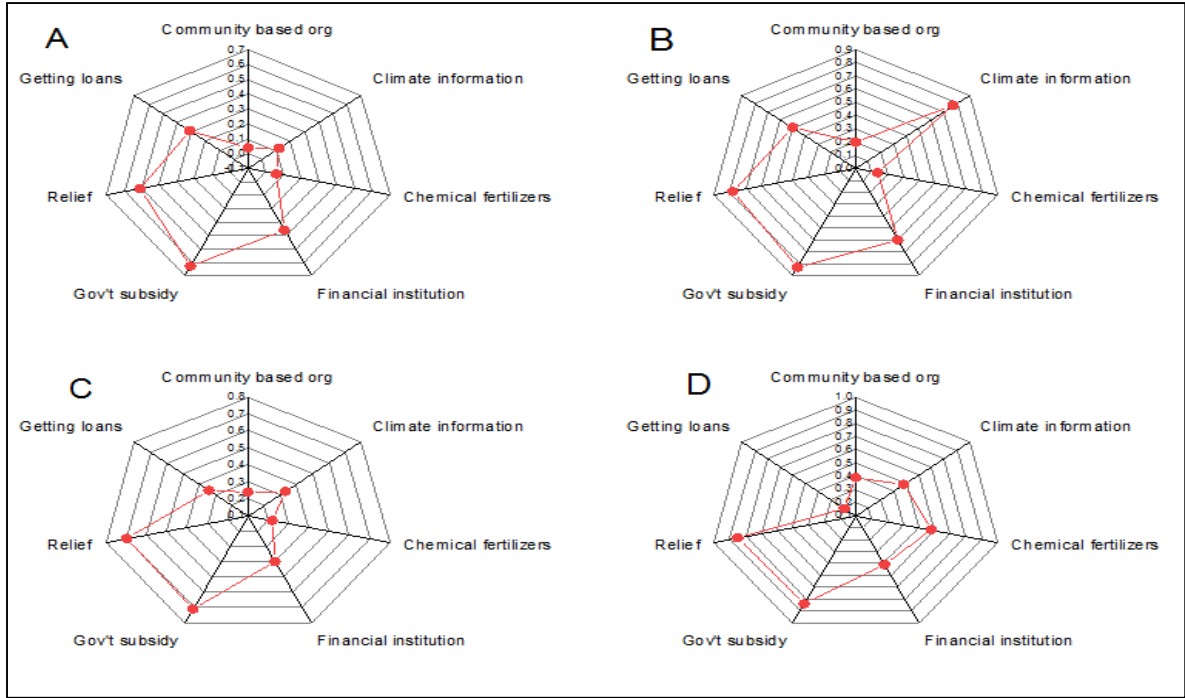

**Figure 3.** Social capital vulnerability index for (**A**) Arjo district, (**B**) Bako Tibe district, (**C**) Chewaka, and (**D**) Sekoru districts.

The existence of community-based organizations (0.04), access to chemical fertilizers (0.06), access to climate information (0.12), a culture of obtaining loans from friends (0.31), and access to financial information (0.36) makes the Jimma Arjo district less vulnerable to the impact of climate change. In the Chewaka district, access to government subsidies (0.71) and access to disaster relief assistance (0.70) recorded higher indices compared to chemical fertilizers (0.22), community-based organizations (0.24), access to climate information (0.33), obtaining loans from friends (0.34), and access to financial institutions (0.40). Accessibility to social networks, social claims, and associations can increase community resilience to climate change impacts [41]. Even though the overall social vulnerability for Bako Tibe and Sekoru was identical (0.55), there was a high disparity among the subcomponents of social vulnerability. Both the Bako Tibe and Sekoru districts experienced low levels of community access to government subsidies and disaster relief assistance. Social capital plays a crucial role in enhancing the public adaptive capacity to bounce back depleted resources [92].

### 3.3.3. Financial Capital

The third main component was financial capital, which included seven sub-components: access to banking services, use of microcredit services, use of micro saving services, borrowing from financial institutions, ability to purchase food in the event of crop loss, off-farm income, and diversification of household income (Table 3). The results show that there is great variation among the three districts. For instance, microcredit services (0.83), microcredit savings (0.83), and the culture of borrowing from financial institutions (0.79) increased the vulnerability level of the farming communities in the Sekoru district. Limited sources of income contribute to community exposure to crises during climate shocks [42]. This means that a large number of households in the district were reluctant to use microcredit services and did not benefit from the existing financial institutions due to religious influence. However, the government encourages local communities to take advantage of existing microcredit services with low interest to increase their livelihoods. These problems are reported by the key informant interviews. A large proportion of Islamic religions are not interested in using microcredit services and saving because they perceive that all microcredit services have interest. The majority of Islamic religions are more interested in using interest-free microfinance, which is based on the Shariah profit loss-sharing mechanism [94].

**Table 3.** Financial capital vulnerability index of the Jimma Arjo, Bako Tibe, Chewak and Sekoru districts.

| Indicators of Household Vulnerability Index | Composite Index | | | |
| --- | --- | --- | --- | --- |
| | Jimma Arjo | Bako Tibe | Chewaka | Sekoru |
| Use of bank services | 0.62 | 0.73 | 0.40 | 0.51 |
| Use of microcredit services | 0.61 | 0.44 | 0.40 | 0.83 |
| Use of micro-saving services | 0.60 | 0.46 | 0.45 | 0.83 |
| Borrow from financial organizations in the past | 0.38 | 0.46 | 0.66 | 0.79 |
| Ability to purchase food in case of crop loss | 0.21 | 0.37 | 0.73 | 0.55 |
| Off-farm income generation mechanisms | 0.69 | 0.56 | 0.79 | 0.55 |
| Household income diversification | 0.50 | 0.62 | 0.63 | 0.51 |
| Financial capital vulnerability index | 0.52 | 0.52 | 0.58 | 0.65 |

In the Chewaka district, off-farm income (0.79), ability to purchase food in case of crop failure (0.73), and borrowing from the financial organizations (0.66) recorded higher index scores among the seven identified financial capital that influenced the vulnerability level of the household. In the Bako Tibe district, the use of bank services (0.73) and income diversification (0.62) experienced higher index scores, while off-farm income (0.69) and use of bank services (0.62) recorded the highest index scores in the Jimma Arjo district. According to Dunford et al. [96], financial capital includes household income and savings.

The Jimma Arjo and Bako Tibe districts had the same overall vulnerability index (0.52). The availability of credit services plays an important role in poverty alleviation [97]. Due to low financial capital, vulnerable groups were not able to afford the rising costs of goods [92,98].

### 3.3.4. Physical Capital

Physical capital was the fourth main component, which includes eight subcomponents: household land assets, cultivated farmland in hectares >1.5, use of solar energy for cooking, use of agricultural machinery, access to modern irrigation systems, access to health centers <1 km, access to electricity for cooking, and access to road transportation facilities (Table 4). The results show that four out of eight physical capital types, access to electricity, use of agricultural farm machinery, use of solar energy, and utilization of modern irrigation facilities, scored higher index values. In contrast, in the areas of household cultivated farmland (<1.5 ha), access to health centers (<1 km), household land assets, and access to road transport scored lower index values. Long distances to health facilities can expose people to diseases and health hazards that can affect food security and the well-being of households.

**Table 4.** Physical capital vulnerability index of the Jimma Arjo, Bako Tibe, Chewaka and Sekoru districts.

| Indicators of Household Vulnerability Index | Composite Index | | | |
|---|---|---|---|---|
| | Jimma Arjo | Bako Tibe | Chewaka | Sekoru |
| Household land assets | 0.17 | 0.18 | 0.42 | 0.19 |
| Cultivated farmland in hectare in hectare <1.5 | 0.52 | 0.71 | 0.41 | 0.59 |
| Use of solar energy for cooking | 0.85 | 0.90 | 0.92 | 0.89 |
| Use of agricultural farm machinery | 0.95 | 0.91 | 0.88 | 0.87 |
| Modern irrigation infrastructure | 0.78 | 0.94 | 0.85 | 0.90 |
| Access to health lefts within <1 km | 0.89 | 0.56 | 0.18 | 0.47 |
| Access to electricity for cooking | 0.96 | 0.96 | 0.79 | 0.93 |
| Access to road transport services | 0.42 | 0.17 | 0.53 | 0.07 |
| Physical capital vulnerability index | 0.69 | 0.67 | 0.62 | 0.61 |

Jimma Arjo was the most vulnerable (0.67), followed by Bako (0.67) and Chewaka (0.62), while Sekoru (0.61) was relatively less vulnerable than other districts in terms of physical capital. Jimma Arjo was the most vulnerable due to limited access to electricity (0.96), use of agricultural farm machinery (0.95), access to health centers (0.89), and use of solar energy (0.85). The physical capital of Bako Tibe is lower than that of Chewaka and Sekoru districts due to access to electricity (0.96), use of modern irrigation infrastructure (0.94), use of agricultural farm machinery (0.91), and use of solar energy (0.90).

Access to health facilities (0.18) had the lowest index value in Chewaka, while access to road facilities (0.07) had the lowest indicator score in the Sekoru district. Jimma Arjo's household land assets had the lowest indicator score (0.17), while access to road transport had the lowest score (0.17). The scores for cultivated farmland in hectares (0.41) and household land assets (0.42) at Chewaka, and access to road transport (0.42) and cultivated farmland in hectares (0.52) at Jimma Arjo have lower values, indicating that the household has some resources. Households with good access to physical capital have better livelihood strategies than those without [99].

### 3.3.5. Human Capital

Human capital is a key indicator of household vulnerability to climate change. This study uses six subcomponents of human capital, namely the education status of the household, knowledge of crop varieties, knowledge of improved livestock varieties, household size, household dependency ratio, and household head; these were used to assess the existing human capital. The household demographic factor is one of the key determinants of food security [100]. Human capital includes human knowledge, skills, and capacity to survive during climate shocks [41]. According to interviews with key informants, farming

communities have access to improved crop varieties, but the supply and demand are not balanced. Some people are unable to obtain improved crop varieties on time from offices of agriculture and natural resources. Diversification of plant varieties is one of the techniques to increase the resilience of crop damage to climate change or extremes. This is because different crops have different resilience to climate shocks.

Two of the six human capital types, (1) knowledge of crop varieties, and (2) male household heads, scored the lowest index values across the four districts. In contrast, four subcomponents, namely dependency ratio, knowledge of improved livestock varieties, household size, and household education status, scored higher index values in four districts (Figure 4). Regarding livestock breeds, some households benefited from artificial insemination by veterinarians, but access to improved livestock breeds is limited across the study area compared to improved crop varieties. Educated households are more likely to be more aware of climate change and adopt new technologies to minimize climate change-related risks [82,101,102].

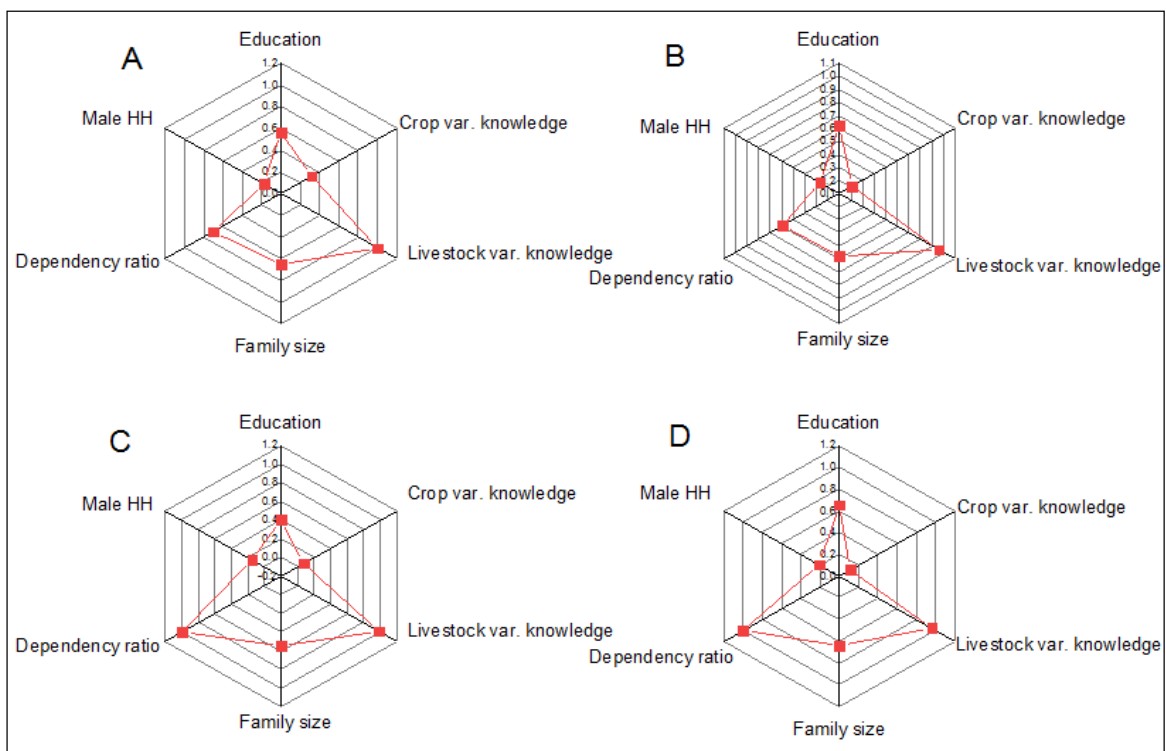

**Figure 4.** Human capital vulnerability index for (**A**) Arjo district, (**B**) Bako Tibe district, (**C**) Chewaka, and (**D**) Sekoru district.

The results show that there is great variation in the dependency ratio across the study areas. For instance, the dependency ratios for Bako Tibe, Jimma Arjo, Sekoru, and Chewaka were 0.59, 0.70, 1.03, and 1.15, respectively. Compared to the other districts, Bako Tibe has fewer economically inactive family members, which might be due to the lower number of children. On the other hand, the higher dependency ratio in Chewaka and Sekoru districts might be due to the high human fertility rate, which was influenced by cultural and religious beliefs to use family planning. Having many children is encouraged, and limiting the number of children is a sin in the Islamic religion [101]. Therefore, the Islamic religion discourages the use of family planning to limit the number of children [102].

A higher dependency ratio increases the vulnerability of the household member due to less capacity to afford food prices [76]. The score values for education status were 0.40, 0.56, 0.62, and 0.65 in Chewaka, Jimma Arjo, Bako Tibe, and Sekoru districts, respectively. Educated households have the capacity to deal with climate change and find alternative options [23]. Therefore, the high dependency ratio, limited knowledge

of improved livestock varieties, large household size, and low educational status of the household heads are the main reasons for household vulnerability to the effects of climate change in the study area.

### 3.4. Household Vulnerability Based on Five Indicators

The radar pectoral of the five major capitals is presented in Figure 5. The results show that Jimma Arjo has relatively good social capital (0.29) and natural capital (0.33). The lowest vulnerability to climate change in the Jimma Arjo district was due to the existence of community-based organization (0.04), access to chemical fertilizers (0.06), less exposure to rainfall deficit (0.08), less exposure to extreme cold (0.08), and climate suitability of agriculture (0.09).

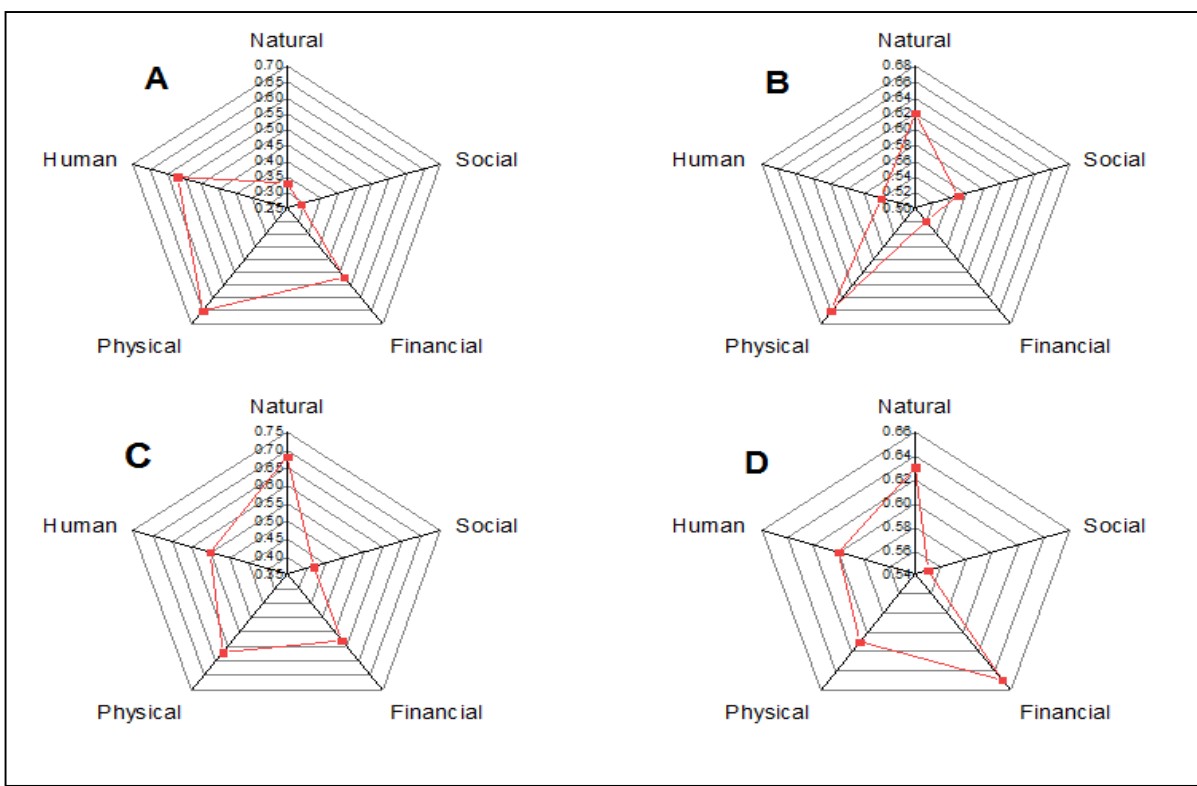

**Figure 5.** Radar pictorial presentation of overall household vulnerability indices based on five major capitals for (**A**) Arjo district, (**B**) Bako Tibe district, (**C**) Chewaka district, and (**D**) Sekoru district.

The results of the physical capital index revealed that Jimma Arjo scored the highest values (0.69), followed by Bako Tibe (0.67), while Chewaka and Sekoru districts scored 0.62 and 0.61, respectively. Regarding the human capital index, Sekoru and Jimma Arjo districts scored 0.60, while Jimma Arjo scored 0.57. Bako Tibe (0.54) and Chewaka (0.55) scored comparable values. On the other hand, Sekoru district had a lower financial capital (0.65), while Chewaka districts had a moderate social vulnerability index (0.58), and the Jimma Arjo and Bako Tibe districts had a lower social vulnerability index (0.52).

The overall household vulnerability index shows that the highest vulnerability is detected in Sekoru district (0.61), followed by Bako Tibe (0.58) and Chewaka (0.57), while Jimma Arjo district experienced the lowest level of vulnerability (0.48) to climate change impact. Bako Tibe and Chewak have similar overall vulnerabilities, but there are significant differences between the Sekoru and Jimma Arjo districts. A study by [103] found the occurrence of extreme and severe drought in the Sekoru and Jimma Arjo districts, exposing farming communities to the impacts of climate change.

## 4. Conclusions

The cumulative effects of rainfall irregularities and extreme weather events, such as erratic and excess rainfall, exposed the farming communities to the impacts of climate change. The results show that educated households can easily understand the impact of climate change on agricultural production. Educated households also described several adaptation options and their willingness to combat the impacts of climate change. Education is, therefore, a key factor in influencing the household head to adapt to climate change. There are significant links between perceptions of climate change and changes in rainfall and temperature patterns, as well as the occurrence of climate extremes such as droughts and floods. People who can understand changes in rainfall and temperature patterns and the occurrence of climate extremes will be able to recognize climate change impacts and take necessary adaptation measures. The occurrence of early rain and late rain significantly affects agricultural production, and consequently, more than half of the household heads perceive these problems. Irregularity of rainfall is a key problem that significantly affects agricultural production.

The key informants highlight that climate extremes, particularly drought and flood, affect agricultural crops. It is evident that climate change significantly affects the rain-fed dependent agricultural economy, leading to food insecurity. The vulnerability of households is mainly associated with climate change impacts such as changes in rainfall and temperature patterns and the occurrence of droughts and floods. Moreover, the lack of much-needed infrastructure facilities, weak institutional support, and limited access to natural, social, physical, financial, and human capital have increased the vulnerability of communities to the impacts of climate change.

Due to differences in natural, social, physical, financial, and human capital, there are large differences in the extent of household vulnerability to climate change across districts. The government and other nongovernmental organizations can increase the adaptive capacity of farming communities by providing improved varieties of crops and livestock, affordable agricultural inputs, weather information, and enhancing microcredit services and other possible strategies to minimize the vulnerability of the local community to the effects of climate change. As future trends in climate change are fraught with uncertainty, governmental and nongovernmental organizations should establish climate-resilient mechanisms to ensure the sustainability of farmer livelihoods in the region and beyond. People have different beliefs and understandings about their perception of climate change and their vulnerability to climate change impacts. Thus, further studies can be conducted using an unequal weighting approach based on expert judgment or principal component analysis.

**Author Contributions:** Conceptualization, D.O.G., D.K. and W.G.; methodology, D.O.G., D.K. and W.G.; software, D.O.G.; validation, D.O.G.; formal analysis, D.O.G., D.K. and W.G.; investigation, D.O.G., D.K. and W.G.; resources, D.O.G., D.K. and W.G.; data curation, D.O.G.; writing—original draft preparation, D.O.G.; writing—review and editing, D.O.G., D.K. and W.G., visualization, D.O.G.; supervision, D.K. and W.G.; project administration, D.O.G.; funding acquisition, D.O.G. All authors have read and agreed to the published version of the manuscript.

**Funding:** This work was supported by Jimma University College of Agriculture and Veterinary Medicine (JUCAVM). JUCAVM support the First Author by providing a monthly salary and transportation cost for household survey.

**Institutional Review Board Statement:** Not applicable.

**Informed Consent Statement:** Not applicable.

**Data Availability Statement:** The data used for this study is confidential. The data can also be available on request from the corresponding author.

**Acknowledgments:** First of all, the authors acknowledge the farming communities of southwestern parts of Ethiopia for providing the necessary information to carry out this study. Secondly, we acknowledge all stakeholders and key informant interviews for their willingness to respond to the designed questionnaire on public perception and community vulnerability to the impact of climate change in southwestern parts of Ethiopia. Finally, we acknowledge all synonymous reviewers for providing valuable inputs.

**Conflicts of Interest:** The authors declare no conflict of interest.

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
