# Peer review of "Climate Change Perception and Vulnerability Assessment of the Farming Communities in the Southwest Parts of Ethiopia"

_climate, doi:10.3390/cli11090183_

Round 1

Reviewer 1 Report

This study aims to address the existing research and knowledge 90 gaps on community vulnerability to climate change in SW Ethiopia. It is more comprehensive and includes natural, social, financial, physical and human capital to assess household vulnerability to climate change. However, this paper doesn’t look ready for publication for some visible mistakes, which make the paper’s quality low. Here are some comments.

[1] In Title, I suggest “Southwest” to replace “Wettest” since “Southwest” has appeared in the Keywords , Section 1 introduction and Section 2.1 Study area description.

[2] The structure of this paper could be shown in the last part of Introduction.

[3] There are two Section “2.1”,  three Section  “2.1.1.”, three Section 3.1.”. Pay attention to section numbering. Double check the section numbering of the whole paper.

[4] Avoid mistakes and improve the writing quality of this paper.

Author Response

Dear anonymous reviewer, thank you so much for providing valuable comments on our manuscript entitled as ‘’Climate Change Perception and Vulnerability Assessment of the Farming Communities in the Southwest parts of Ethiopia.’’ All comments and suggestions are helpful to enhance the quality of our manuscript.

Per your comments and suggestion, we improved the introduction, the research design, the methods, results and discussions as well as the conclusion section.

Our response to specific comments and suggestions for authors are attached through journal editorial pages.

Reviewer 2 Report

Comments_Climate-2372492: Climate Change Perception and Vulnerability Assessment of the Farming Communities in the Wettest parts of Ethiopia

Recommendation: MINOR Revisions

The subject paper is a nice effort regarding the climate change perception and vulnerability assessment of the farming communities in Ethiopia. The message and idea of the paper are appealing and structured appropriately. The language of the paper needs some sort of improvement, however.

Abstract: It is nicely drafted. However, it is better to mention the sampling technique in this section.

Title: The title is ok.

Introduction: It is well-presented and coherently developed if some typos or grammatical mistakes are handled with just a thorough read by the authors. Moreover, Afro-Asian climatic conditions are pretty similar, so it would aid to literature to cite some recent studies conducted in a highly vulnerable Asian country i.e., Pakistan (e.g., Mahmood et al. 2020; Mahmood et al. 2021).

Methods: This section has been drafted in a good scientific way.

Results and Discussion: I really appreciate the writing style and format of this section. The author can have a look into abovementioned suggested studies (in the introduction section) to second some of his variables’ behavior/findings.

Conclusion: Based on study findings, this section is ok.

1-     Mahmood, N., Arshad, M., Kächele, H., Shahzad, M. F., Ullah, A., and Müller, K. (2020). Fatalism, climate resiliency training and farmers’ adaptation responses: Implications for sustainable rainfed-wheat production in Pakistan. Sustainability, 12(4), 1650

2-     Mahmood, N., Arshad, M., Mehmood, Y., Shahzad, MF, & Kächele, H. (2021). Farmers' perceptions and role of institutional arrangements in climate change adaptation: Insights from rainfed Pakistan. Climate Risk Management, 32, 100288.

Author Response

Dear anonymous reviewer, we acknowledge your scientific comments and recommendation to improve the quality of our manuscript entitled as ‘’Climate Change Perception and Vulnerability Assessment of the Farming Communities in the Southwest parts of Ethiopia.’’ All comments helpful to enhance the quality of our manuscript. We improved the introduction section. The results sections also improved per the comments (See the track changes on the main documents).

Reviewer 3 Report

Comments and suggestions:

  1. The author could develop a framework for vulnerability assessment in the methodology section.
  2. How did the author select the indicators under the five livelihood capitals? The author could present the selected indicators in a table with their sources from the ongoing research communication.
  3. Usually, in vulnerability assessment, some indicators are positively related to vulnerability while others are negatively related. But there is no indication of which of the indicators are positively and negatively related to vulnerability and how they aggregate in the overall vulnerability score.
  4. One of the major limitations of the study is considering the equal weights of all the selected indicators, which is considered bias. Because, in reality, some of the indicators might have influenced the vulnerability more than other factors. The author could adopt unequal weight by expert judgement or principal component analysis to avoid bias. Or, there are some statistical techniques to estimate indicator weights to avoid bias!
  5. The author could adopt some statistical model to identify the most influencing factors of vulnerability across different study locations, which will help policymakers formulate policies to improve the resilience capacity of the community.
  6. I would suggest writing a separate discussion section based on the findings, followed by strong arguments.
  7. The author should also highlight some of the limitations of the study.

Author Response

Dear reviewer, thank you so much for your contribution on our manuscript. All comments are helpful. I confirm you that we addressed your comments and concerns. See the revised manuscript in track change. You can also assess our response in the attached file.

Reviewer 4 Report

Climate Change Perception and Vulnerability Assessment of the Farming Communities in the Wettest parts of Ethiopia The paper is quite interesting but there are many similar studies both for Africa and rest of the world but there is some scope of this work for local relevance. the background and introduction of the topic is nicely portrayed but but the methods and the subsequent findings are deficient one way or the other. As the major goal of the paper is evaluating perception and vulnerability of farmers, the first part is just a description of the people who perceived some aspect to be there or not just in the form of yes or no without considering other options of perception between yes and no as for example, one farmer may high perception or relatively low perception os this could be done in the form of ordered response instead of just two options. there are many confusing statements in the paper as well based on findings. For example the statement 'Concerning the association between crop loss and food insecurity with climate change, the number of perceived respondents was comparable with those who did not perceive climate change (p value <0.001).' below Table 1 but there is no such result to support the conclusion rather contradicts with the study quoted to support the findings in Table 1 as the percentage of people who perceived food insecurity due to climate change is less in comparison with thos who do not perceive. This can further be confirmed from: Sam et al. (2021) Flood vulnerability and food security in eastern India: A threat to the achievement of the Sustainable Development Goals. International Journal of Disaster Risk Reduction, 66, 102589. and Sam et al. 2018. Linking Food Security with Household’s Adaptive Capacity and Drought Risk: Implications for Sustainable Rural Development. Social Indicators Research.  doi.org/10.1007/s11205-018-1925-0.  In addition, instead of looking for the factors affecting perception of the farmers, authors have given a model of binary logistic by considering household awareness level of climate change as dependent variable in Equation 14. This too is questionable as awarness is neither the topic of the paper nor such information has been collected rather just perceptions of particular outcomes as documented in Table 1 are the main outcome of the survey. So, even if this variable could have been used, there is no uniform indicator to be used as binary response variable. This shortcoming is further intensified by not estimating the model despite its mentioning in the methods section. So this is a serious flaw of the work, without which there is no validity of the survey and rest of the analysis on Vulenrability measurment and composite indexation although there is some more information needed to convince reader how those variables/statements were framed to calculate a composite vulnerability index as given in Tables 2-4. The conclusionis are just regulation discussion found in many articles but they are poorly supported empirically by this work. 

English is ok.

Author Response

Dear reviewer, thank you so much for providing valuable comments and inputs on our manuscript. All comments are important to shape our work for scientific communities. 

Reviewer 5 Report

It’s an interesting study, which can contribute to local-level climate risk management significantly (ID: climate-2372492). Overall, the manuscript is written well. However, there are some points in the manuscript that needs to be incorporated. The paper needs the below-mentioned editing and revision before it is considered for publication in the Journal.

1.      In the abstract, information related to five household capitals (natural, social, financial, physical, and human capital) is repeated in lines: 16, 20-21, please revise it.

2.      The introduction needs to be revised. Need more in-depth narration to strengthen your motivation for the study, the applicability of methods, objectives, and significance of the study. I would suggest revising the introduction section with a clear and detailed explanation focusing on the importance of climate change perception, how it contributes to the field of climate change focusing on global aspects, how it’s beneficial to local level climate risk management, why such study is important to your studied area, what are the related past study’s findings, and what are the gaps in existing literature. Importantly, precipitation and temperature are the two pivotal parameters of global climate, so, changes and variability in these two parameters as well as associated extreme events how affect your study area. Discuss it considering both historical and future perspectives.

3.      In section 2.1, please also add a global map indicating your study area position with the existing map. Please add here some recorded information on socioeconomic damages due to major hazards in your study region.

4.      A methodological flowchart would be better to understand the design of the study clearly.

5.      What is the basis of your selection of specific sub-indicators to estimate vulnerability: natural capital (12 sub-indicators), social capital (7 sub-indicators), financial capital (7 sub-indicators), physical capital (8 sub-indicators) and human capital (6 sub-indicators). Is it based on the literature review or key informant interviews? If it is based on the literature review please explain it logically. It would be more acceptable and reasonable if you select these vulnerability indicators based on key informant interviews.

6.      The discussion is limited here. It should highlight the insights and the applicability of your findings/results, the reasoning behind the changes, and justify your findings with previous results.

7.      Please mention some shortcomings in this research; uncertainties involved with data and methods.

8.      The conclusion should be more concise and simplified.

9.      The language of the manuscript needs to be improved.

The language of the manuscript needs to be improved.

Author Response

Dear reviewer, thank you for providing helpful comments. We incorporated all comments. See the revised version of our manuscript as well as our response to your comments.

Round 2

Reviewer 1 Report

The revised version looks better. I can see the authors have made their efforts to address my comments. However, there are still some comments for improving the quality of this paper.

[1] In line 21 “Results on household perceptions of climate change were statistically significant relationship between…”, here “were” could be “showed”.

[2] In line 186 “In this study, the method of triangulation proposed by [63] and used by several authors…”, I suggest adding the author’s name in front of “[63]”.

[3] In line 188 “Ref. [64] demonstrated that…”, delete “Ref.” and add authors’ names in front of “[64]”. Avoid such cases in the whole paper. Always show the authors’ names in front of the citations. It helps the readers to know the references and shows respects to the authors.

Author Response

Thank you so much for checking the revised manuscript. We fully incorporated your comments and suggestion in this final version. All comments are helpful.

Reviewer 3 Report

The overall quality of the manuscript has been improved after revision . The manuscript can be accepted for publication. 

Author Response

Thank you for accepting the revised version.

Reviewer 4 Report

I find some of my comments well taken up but concerning equation four, things are not still in perspective. First of all, Line 235 needs to be corrected for Y`=1 (....of not being perceive climate change), possibly to 'probability of not perceiving climate change' or some other. As authors mention binary logistic model, they give results for probit (as per the caption but the results are not for that model, it only gives Chi-square test of comparison of two groups of respondents, against what is provided in Equation 14. If it is so, what about Xi's (vector of other determinants) βi (the vector of unidentified parameters). So its better to remove this equation and change the caption of Table 1 otherwise the model should be estimated and results presented according to Eq. 14 given the values of parameters with their significance level. Rest is fine for me.

Author Response

Dear reviewer, thank you so much for your good comments and observation. Now, we removed the confusing equation 14. Similarly, we modified the caption of Table 1 as household perceptions of climate change.

Reviewer 5 Report

The Author satisfactorily addressed all the suggested comments. It can consider now to be published.

Author Response

Thank you for recommending our manuscript for publication.